# Transmission of natural scene images through a multimode fibre

Piergiorgio Caramazza[1], Oisín Moran[2], Roderick Murray-Smith [2] & Daniele Faccio [1]

The optical transport of images through a multimode fibre remains an outstanding challenge with applications ranging from optical communications to neuro-imaging. State of the art approaches either involve measurement and control of the full complex field transmitted through the fibre or, more recently, training of artificial neural networks that however, are typically limited to image classes belong to the same class as the training data set. Here we implement a method that statistically reconstructs the inverse transformation matrix for the fibre. We demonstrate imaging at high frame rates, high resolutions and in full colour of natural scenes, thus demonstrating general-purpose imaging capability. Real-time imaging over long fibre lengths opens alternative routes to exploitation for example for secure communication systems, novel remote imaging devices, quantum state control processing and endoscopy.

---

[1] School of Physics and Astronomy, University of Glasgow, Glasgow G12 8QQ, UK. [2] School of Computing Science, University of Glasgow, Glasgow G12 8QQ, UK. Correspondence and requests for materials should be addressed to R.M-S. (email: roderick.murray-smith@glasgow.ac.uk) or to D.F. (email: daniele.faccio@glasgow.ac.uk)

Optical fibres form the backbone of the Internet and are a key technology in modern society[1]. The vast majority of these fibres are 'single mode', i.e. they can transmit only one single, roughly Gaussian-shaped beam profile, corresponding to the so-called fundamental mode of the fibre[2]. It is therefore impossible to directly transmit images through an optical fibre: any attempt to do so simply results in transmission of the one single allowed mode and therefore the detection at the output of this (Gaussian-shaped) mode, with all other information of the image completely lost at the fibre input. One possibility to circumvent this limitation is to resort to an array or bundle of single-mode optical fibres, each one transmitting the information of a single pixel in the output image. However, this quickly leads to fibre-bundled cables that are relatively thick and not optimal for applications, such as endoscopic or neurological imaging, where the fibre bundle is inserted inside a body[3].

Another option is to resort to multimode fibres, i.e. fibres that, due to a larger core diameter, can carry many optical modes that will have more complex shapes than the fundamental mode and may encode image information[4]. For example, a typical 100-μm core diameter fibre might carry around 10,000 modes and could in principle transmit an image with roughly the same number of pixels. However, in these fibres, each of these individual modes propagates at a slightly different velocity, thus leading to an amplitude and phase mixing of the image, as this propagates along the fibre[5]. The image at the fibre output therefore appears as a random array of bright and dark spots, referred to as a speckle pattern. This effect cannot be avoided and completely scrambles and destroys the input image. Full a priori knowledge of the input image and fibre details could allow to numerically model the optical propagation[6], reconstruct the transmission matrix and then unscramble the output data, but in practice, this can be extremely hard. Methods have been developed that allow to shape the input beam profile, so as to focus the output field into a single spot that can then be scanned[7–11], with an emphasis on endoscopy[12–15]. Notwithstanding this notable progress, the development of a viable method that allows to unscramble the speckle patterns and thus retrieve high-resolution, general image information in real time, is an open challenge.

One promising route in this direction is based on the complete characterisation of the optical fibre in the form of a measurement of its transmission matrix[4,12,14]. This matrix connects certain orthogonal modes at the fibre input to the fibre output and can therefore, once known, be used to invert the speckle pattern back into the original image. This approach requires measurements of the full complex (amplitude and phase) profile of a large subset of modes and has been shown to work over fibre lengths of 0.3–1 m.

Other approaches pioneered by Takagi et al.[16], have recently been proposed, that used artificial neural networks (ANNs) using deep-learning encoders to infer images from the speckle patterns without any need for an a priori mathematical model of the fibre[17–19]. These have used multi-layer convolutional ANNs and have shown that it is possible to reconstruct handwritten digits from the MNIST database[20] that consists of patterns with 28 × 28-pixel resolution: training of the network and testing of its reconstruction abilities are both performed on digits from the same database. ANNs have also been shown to allow, for example, to focus a beam through a thin scattering medium (as well as through a multimode fibre) in ref. [21], where both single-layer and multi-layer real-valued neural networks have been implemented. When used for imaging, these approaches are mostly expected to work for classes of objects that belong to the same class used for the training, as explicitly pointed out by Psaltis et al.[17]. First steps towards generic imaging have been made in ref. [18]: these approaches present a very promising route forward if the suitability for general-purpose imaging applications can be addressed.

We have developed an approach that allows us to transmit and reconstruct images of natural scenes at high resolution and frame rates. Our method resorts to building an approximate model of the inverse of a complex-valued, intensity transmission matrix of the optical fibre. This approach does not require the use of deep (multi-layer) ANNs and enables the full reconstruction of detailed images. Full colour images and videos can be recorded at 20 fps and could be scaled up to kfps.

## Results

**Experiments.** The experimental layout is shown in Fig. 1a. We use an SLM (maximum frame rate of 20 Hz) to impart greyscale (100 greyscale levels) intensity images onto a continuous-wave laser beam (532-nm wavelength). This image is then coupled into a multimode fibre (step index core, core diameter 105 μm, fibre lengths of 1 and 10 m, ~9000 propagating optical modes and image spot size at fibre input is ~2 μm) and then coupled out using identical objectives for the fibre input and output (focal length, $f = 34$ mm, NA = 0.26). The speckle pattern at the fibre output (near field) is imaged onto a CMOS camera at $350 \times 350$-pixel resolution.

The goal is to transmit 'natural scenes', i.e. photographs of everyday-life scenes. The importance of this choice lies in the significant additional complexity of natural scenes when compared with, e.g. MNIST-database digits or other simple geometric features. As sample images, we use a selection of 50,000 images from the Imagenet database[22], sized at $92 \times 92$ pixels, so as to have less pixels than optical fibre modes. These images have been selected randomly from the Imagenet database while looking for almost square images, so as to facilitate projection into the fibre. The output speckle patterns with amplitude distribution, $x$ (i.e. $x$ is the square root of the measured speckle intensity patterns), together with the knowledge of the image (intensity distribution, $I$) that generated each speckle pattern, are used in the algorithm

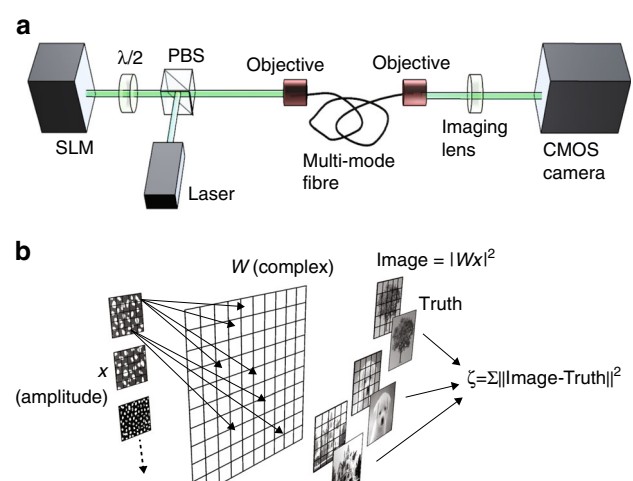

**Fig. 1** Experimental layout. **a** An SLM combined with a polarising beam splitter (PBS) and a half-wave plate ($\lambda/2$) is used to imprint intensity images onto a laser beam that is then coupled into a multimode fibre. The fibre output is collected with a lens and recorded on a CMOS camera. **b** A schematic overview of the computational processing steps of the inversion process: output speckle data, $x$ from a series of images (in our experiments, 50,000 images from the ImageNet database[22]) are fully connected to a complex matrix, $W$ which provides an output image $I = |Wx|^2$. This image is compared with the actual original image (ground truth) through a cost function: the total cost $\zeta$ is then back-propagated to W and the process is repeated for a fixed number of loops (epochs), ensuring minimisation of $\zeta$

described below to approximate the inverse of a complex transmission matrix, $W$. This matrix is then used to retrieve images that were not part of the sample dataset from intensity measurements of their output speckle patterns, $I = |Wx|^2$. Examples are images and videos from the Muybridge collection, such as a running horse, a jumping cat and a flying parrot. We also tested the imaging on videos of a rotating Earth and Jupiter. Both of these are in full colour, obtained by projecting and then recombining the R, G and B channels independently.

**Image reconstruction**. There are two possible approaches to reconstructing an image from a speckle pattern. The first is to attempt to build a forward model that describes how the images or optical modes propagate down the fibre and then invert this or, the approach followed here, one can try to directly construct an approximation of the inverse model. This is achieved statistically (i.e. by employing data from many images) through a single, fully connected complex-valued transformation matrix.

**Complex inversion**. A schematic overview of the approach used to reconstruct $W$ is shown in Fig. 1b. The full transmission matrix of an optical fibre is complex valued. This motivates the assumption that $W$ is also complex valued, connecting the input and output of the fibre $I = |Wx|^2$ [23–26]. This also motivates the idea that a deep-learning ANN approach is not required here. We only measure the intensity of the speckle pattern, from which we take the amplitude $x$ (square root of the intensity) and therefore represent $x$ with amplitude only, and zero phase. The values of $x$ are passed to a fully connected ('dense') complex matrix −equivalent to multiplication by the complex-valued matrix $W$ (arrows in Fig. 1b) that shows some of these connections as an example. An image is then obtained as $I = |Wx|^2$. We calculate the derivatives $d\zeta/dw_{ij}$ of the cost function $\zeta$, with respect to the $i$, $j$th element of $W$. We then apply a stochastic gradient descent approach to make small changes to $W$ that reduce the cost function, and the process is repeated for a fixed number of loops (epochs), ensuring convergence of $\zeta$ to a minimum value. The complex-weighted inversion was implemented as a novel layer with Keras[27] and TensorFlow[28] (the code is provided in Supplementary Note 4). The final $W$, constructed from a database of 50,000 images, can then be used to obtain an estimate of the ground truth image for all future transmitted data corresponding to images not used as part of the training and indeed, even transmitted at a completely different time (e.g. several days) after the training is completed.

**Experimental results**. In Fig. 2, we show a first set of results obtained from data transmitted through a 1-m-long fibre (wrapped in a loose coil on the table), by applying the estimated $W$ to a series of videos that do not form part of the ImageNet database and are significantly diverse, in order to demonstrate the robustness of our approach. These videos are taken from the Muybridge recordings from the 1870s that marked the historically important breakthrough of the first ever high-speed photography images. The running horse in Fig. 2a is probably the most iconic of the Muybridge videos, but the others, a jumping cat in Fig. 2b, a flying parrot in Fig. 2c and a punching boxer in Fig. 2d provide a broad scene variability and all show good image reconstruction over the full greyscale spectrum, as opposed to the binary black/white MNIST images often used in previous work. We also note that the MNIST database, given the limited set of symbols, will tend to create what is essentially a simple classification system, with a decoder to generate the associated image. True imaging capability should therefore demonstrate functionality beyond this dataset, as shown in Fig. 2 (see Supplementary Fig. 3 for further

examples). Under each reconstructed image in Fig. 2, we also give a quantitative measure of the reconstruction quality based on the 'structural similarity index' (SSIM) and the 'Pearson correlation coefficient' (see Supplementary Note 2). These coefficients measure the similarity between the ground truth and retrieved images with a maximum value of 1 (indicating image identity).

We also observed no degradation in the video quality even when the data were transmitted, recorded and reconstructed more than 48 h after transmission of the original first set of 'training' data had been completed (see Supplementary Note 3), thus indicating robustness to subsequent environmental changes, such as temperature fluctuations (of the order of a few degrees) and vibrations (the setup is not placed on a vibration-isolated table).

In Fig. 3, we show examples of full colour video transmission of a rotating Jupiter and a rotating Earth. Each individual R, G and B channel was transmitted and reconstructed separately and then recombined. We note that the same matrix $W$ obtained for greyscale images is used for all three R, G, B channels when imaging in full colour mode. Subtle features, such as the Red Spot on Jupiter or slightly lighter areas in the Northern region of Africa (roughly corresponding to the Nile delta region in Egypt), can be observed in the reconstructed images (other examples are shown in Supplementary Fig. 2).

Tests were also performed to investigate the role of the class of images used for retrieving $W$. The images in Fig. 4 were downsampled to $28 \times 28$ pixels in order to simplify the problem and demonstrate that, if desired, one may also reconstruct the inversion matrix $W$ with a completely 'agnostic' approach, i.e. with no prior assumption on the images. This is obtained by using 50,000 completely random greyscale images. As can be seen, this completely agnostic approach is still able to correctly reconstruct the images, although with a clear loss of quality. We noticed a good insensitivity to fibre length (as already pointed out by Psaltis et al.[17]) and an improvement of image quality with increasing the number of random images used for the $W$ matrix retrieval, although GPU RAM limitations did not allow us to investigate this further.

We also noted that changing the size of the focused image at the fibre input significantly impacts the final reconstruction. By placing a telescope after the SLM so as to rescale the image at the focusing objective input, we noticed a significant increase of the final image quality with increasing the size at the focusing objective input pupil (corresponding to an increasing effective NA, i.e., to an increasing angular spread at the fibre input). This was also accompanied by a clear decrease in the average speckle spot size at the fibre output, indeed indicating the excitation of higher spatial frequency modes (see Supplementary Note 1 and Supplementary Fig. 1).

A matter of concern in many studies is the robustness to changes in the fibre configuration. A change in the fibre geometry (e.g. by bending the fibre) will lead to a different propagation of the individual modes, which ultimately leads to a different output speckle pattern. Without precise knowledge of how the fibre has been changed, it is not possible therefore to reconstruct the image using the inversion matrix $W$ from a different configuration[14]. A recent solution has been proposed by using specially designed fibres that have a parabolic refractive index profile in the core[29]. For a long-range transmission system, such a solution might be appropriate under the assumption that a long-haul fibre would remain in a relatively fixed position over time.

In future realisations, we expect that a combination of fibre design, position classification and/or extensive training over fibre configurations, will allow to efficiently remove this last obstacle that for the time being, is beyond the scope of this work that is aimed at demonstrating that high pixel-density, colour images

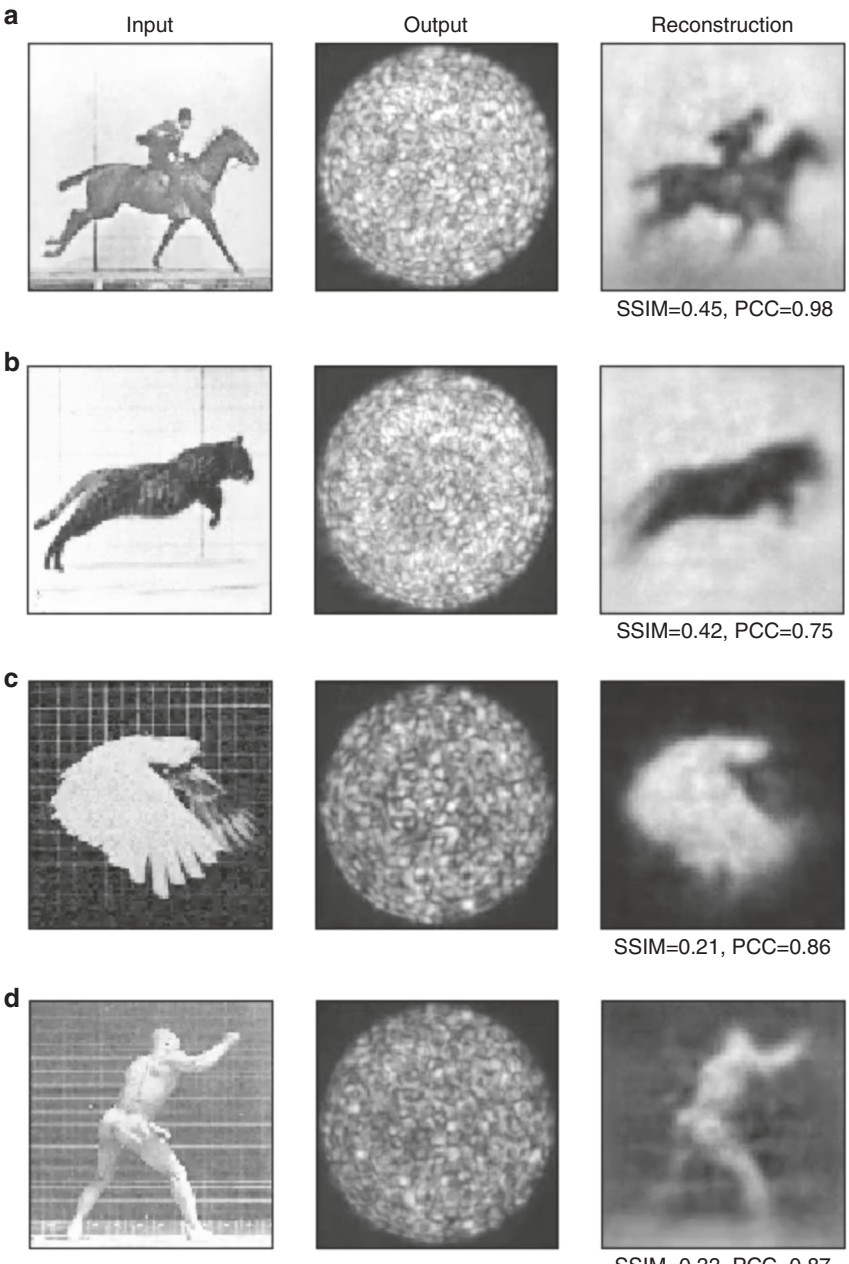

**Fig. 2** Reconstruction of Muybridge videos for a 1-m-long fibre. Individual frames of video data taken from the Muybridge collection: **a** running horse, **b** jumping cat, **c** flying parrot and **d** punching boxer. All videos are greyscale, scaled to 92 × 92 pixels and transmitted at four frames per second. The first column shows the original input video. The second column shows the speckle patterns ($x$) measured at the fibre output. The full videos are available in the supplementary information (Supplementary Movie 1). SSIM and PCC indicate the structural similarity index and Pearson correlation coefficient that quantify the quality of the reconstruction (see Supplementary Note 2)

can be transmitted efficiently through a static fibre and at video frame rates.

## Discussion

Imaging through a single multimode fibre is an outstanding challenge and has so far been limited to relatively low frame rates, small image sizes or short fibre lengths. Imaging of natural scenes increases this challenge further, as it ideally entails video frame rates, colour detail and sufficient image resolution to allow identification of the scene details. The technique developed here is based on a physically informed model of the imaging system that retrieves an approximation to the full transmission matrix. In this

sense, our approach sits somewhere between the techniques devised for reconstructing the actual transmission matrix and deep-learning approaches that do not make any explicit assumptions on the system. This allows for efficient video and data transmission through fibres and promises applications beyond endoscopic imaging, such as direct video or multimode data transmission over long fibres for communication systems and fibre sensing by, for example, exploiting the image sensitivity to changes along the fibre length. Moreover, one could also exploit the intrinsic random nature of the multimode coupling and output speckle patterns to securely encode and authenticate data as proposed recently[30,31], where the transmission fibre itself

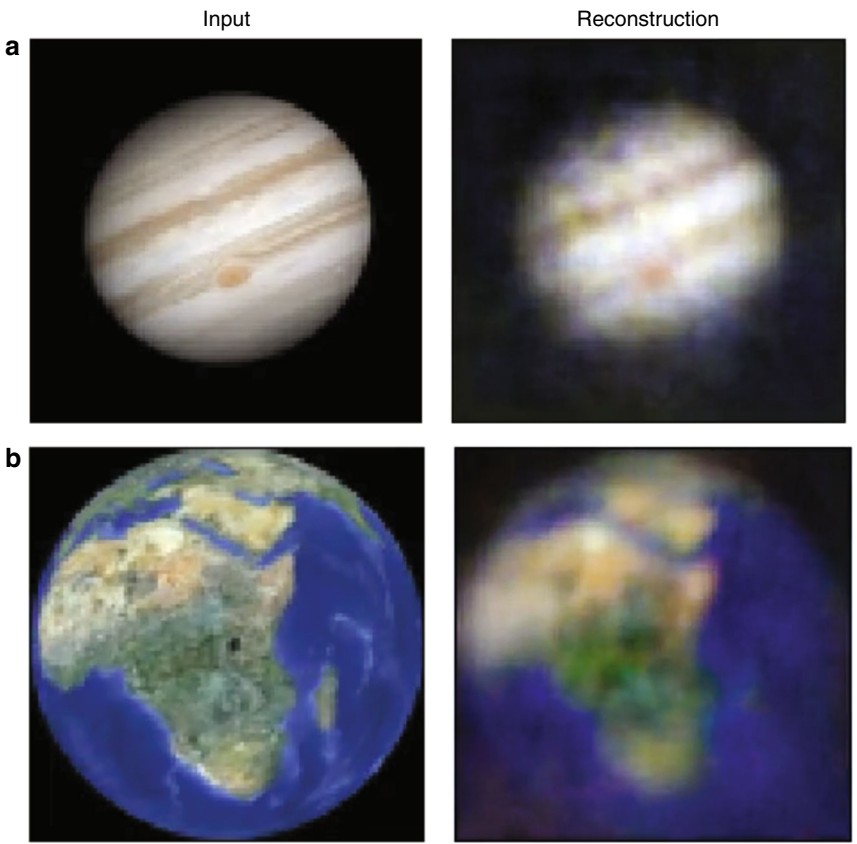

**Fig. 3** Full colour results for a 1-m-long fibre. Individual frames of video data in full colour of a rotating Jupiter (credit: Damian Peach) (**a**) and Earth (**b**)

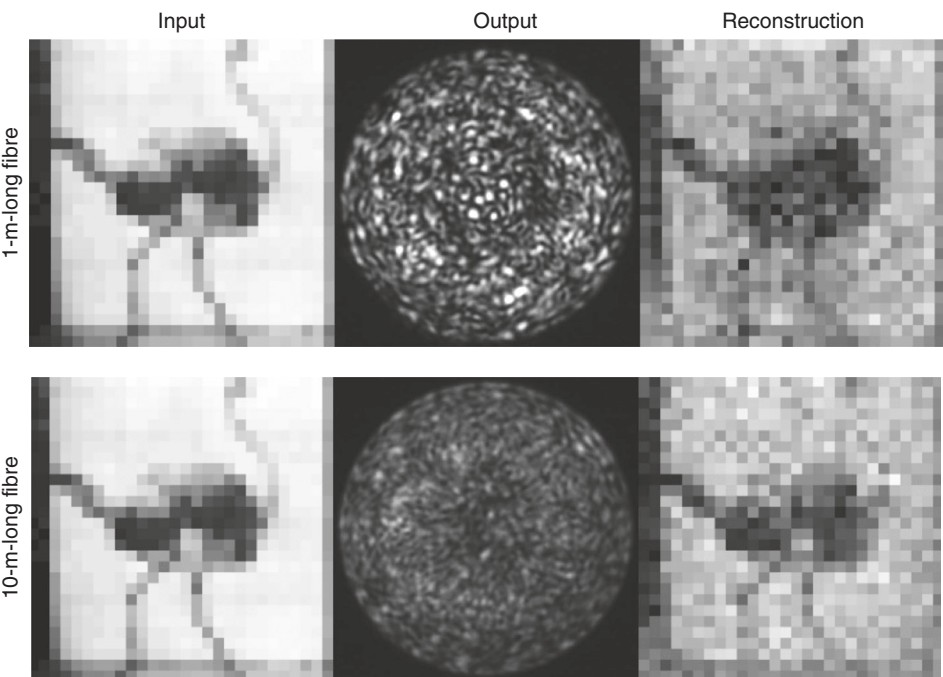

**Fig. 4** Image reconstruction of an ostrich taken from the Muybridge collection after transmission through a 1- and 10-m-long fibre. The inversion matrix was constructed using only random greyscale patterns

would play the role of the encoding medium, with possible extensions also to the control of quantum states for quantum sensing and simulation[32].

**Data availability**

An example of the code is provided along with data for 1 m at https://doi.org/10.5525/gla.researchdata.751.

## Code availability
The code has been presented and explained in Supplementary Note 4: Software. Furthermore, a code example is available at the DOI link.

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

## Acknowledgements
D.F. and R.M.S. acknowledge financial support from EPSRC (UK, grant no. EP/M01326X/1).

## Author contributions
P.C. prepared the experimental setup and performed the experiments. R.M-S proposed the complex-weighted network model, and O.M. and R.M-S implemented and tested the artificial neural network. All authors discussed the results and contributed to the writing of the paper. D.F. conceived and led the project.

## Additional information

**Competing interests:** The authors declare no competing interests.

