## [Peer Review File · Nature Communications]

Reviewers' comments:

Reviewer #1 (Remarks to the Author):

The authors present the transmission of colored images through a multimode fiber by a digital technique that lies between a transmission matrix approach and a deep learning approach.

The manuscript is clearly written.

The authors mention in the abstract and in the main text that previous work on reconstruction with artificial neural networks work only for classes of objects that belong to the same class used for training. This is incorrect as reference [19] demonstrate transfer learning ie. Reconstruction of objects which do not belong to the same class of objects used for training (a picture of a house, heart is reconstructed when the system is trained with letters of the alphabet). Hence the author should correct these statements.

The SLM imaged onto the fiber facet has a spot size of 2 μm according to the authors. With a NA of 0.2 and wavelength of 0.532 nm, how can the SLM image be 2 μm ?

The authors should reference the work of the group of Seelig (arXiv:1805.05602v3). It is quite relevant because they use a single layer neural network which is a complex matrix multiplication followed by a non-linear function. Their work is in a scattering medium and not in a multimode fiber. However because of the speckle pattern output in both cases, this previous study is very relevant.

Could the author discuss more on the choice of images chosen for training.

Could the author elaborate on the reason why their method should converge at all ? A complex matrix relates the amplitude of the field, not the intensity and thus a form of non linear inverse problem need to be solved.

Why a smaller size image is used to compare the transmission performance on a 1 m and 10 m fiber ?

Could the author provide a quantification of the image reconstruction fidelity

Minor points:

Typos: "Examples are images an videos" "Examples are images and videos".

The Arxiv paper cited [19] has been published in a peer reviewed journal and the latter should be referenced.

The manuscript shows surprisingly good results for transmitting images of natural scenes especially that only intensity image encoding and detection was performed. This work should be of interest to the fiber community as a whole.

Reviewer #2 (Remarks to the Author):

This paper by Camarazza et al reports on training a neural network to transmit images through a multimode fiber. There has been recently a number of related works (probably parallel to this one), on using neural networks to classify or reconstruct images through MMF. The work reported here is very similar, but add a number of things that makes it particularly interesting and efficient.

The main one is to use as a neural network not a « black box » deep neural network, but a « simple » unique layer of large size and with a lot of complex valued connections that are then trained. This approach seems very efficient and allow very good image reconstruction. The main claim is that it allows to approximate the inverse transform of a MMF more efficiently than a DNN, and indeed this seems to be the case. By training on a large database (imagenet), reconstruction of images is

demonstrated, interestingly not only on the database itself but also on « new » image unrelated to it. Even learning with «an « agnostic » basis (random inputs) works very well.

The overall reconstruction results are very impressive and convincing, and much better than the state of the art. Still, the conceptual novelty, compared to ref 17-20 is somewhat limited. However, the superior reconstruction capability and the elegant model makes me confident the work is going to have an impact on the community. my recommendation for Nature Communication is therefore (moderately) positive.

The paper is very clearly written, the code is provided, which is very good, my only suggestion to improve the manuscript is to somehow try to be more quantitative on the stability issue: currently the degradation of the reconstruction with time is only very qualitative (paragraph at the end of page 3). It would be nice to give more quantitative data such as correlation of the speckle over time and reconstruction capability. These sensitivity to bending and decorrelation have been studied quite extensively in ref 18 and 20 and it would be nice to see how this method compare.

Reviewer #1 (Remarks to the Author):

1) The authors mention in the abstract and in the main text that previous work on reconstruction with artificial neural networks work only for classes of objects that belong to the same class used for training. This is incorrect as reference [19] demonstrate transfer learning ie. Reconstruction of objects which do not belong to the same class of objects used for training (a picture of a house, heart is reconstructed when the system is trained with letters of the alphabet). Hence the author should correct these statements.

REPLY: we agree with the referee that although we have gone significantly further in terms of generalising, that [19] does show the first attempt to demonstrate transfer learning on a smaller scale and have modified the text to clearly acknowledge the earlier work.

The SLM imaged onto the fiber facet has a spot size of 2 μm according to the authors. With a NA of 0.2 and wavelength of 0.532 nm, how can the SLM image be 2 μm ?

REPLY: This is our fault for creating some confusion with what we were showing and the labelling of the figures in the supplementary information. We have now fixed this. Rather than refer to the focus size of a Gaussian beam illuminating the objective we now refer to the more meaningful actual NA with which the image is focused. The NA of the objective is of course fixed, but the effective NA with which the image is focused depends on the size of the image at the objective input pupil. This NA determines the focusing angle of k-vectors of the image on the fibre input which in turn determines the range of the excited modes in the fibre (lower k-vector images will excite lower order modes, but the same image that is focused with higher k-vector content on the fibre input will excite higher order modes. Given that that higher order (higher k-vector) modes carry higher detail information in the image, this immediately explains why increasing the effective NA leads to an increase in the retrieved image detail, as seen in the experiments. It is therefore this effective NA that underpins the physics of the imaging process and this is the quantity that we now report in the supplementary (together with the above explanation). Changes explaining this have been added to the SM file.

The authors should reference the work of the group of Seelig (arXiv:1805.05602v3). It is quite relevant because they use a single layer neural network which is a complex matrix multiplication followed by a non-linear function. Their work is in a scattering medium and not in a multimode fiber. However because of the speckle pattern output in both cases, this previous study is very relevant.

REPLY: we agree with the referee. We had already cited this paper, we now explicitly point out the similarities.

Could the author discuss more on the choice of images chosen for training.

REPLY: the images were chosen at "random", i.e. by an algorithm that simply looked for images that had a square aspect ratio, without any selection on the actual content. This is now explained in the supplementary/methods.

Could the author elaborate on the reason why their method should converge at all ? A complex matrix relates the amplitude of the field, not the intensity and thus a form of non linear inverse problem need to be solved.

We thank the reviewer for highlighting an issue with our description. We have updated the description and paper to make clear that we are using the "amplitudes", i.e. the sqrt of the measured quantities and retrieve a W, such that $I=|W \cdot \sqrt{x}|^2$. We have also modified the figures so that these now show the results from training with amplitudes rather than with intensities.

Why a smaller size image is used to compare the transmission performance on a 1 m and 10 m fiber ?

REPLY: there was no particular reason for this other than we simply had various sets of data but only lower resolution sets that were identical on both 1 m and 10 m.

Could the author provide a quantification of the image reconstruction fidelity

REPLY: We have now included an quantitative analysis of the image quality using both the Pearson correlation coefficient and the structural similarity index, as explained and overviewed in the supplementary These values are explicitly indicated e.g. in Figure 2 in the main paper and Figure 2 in the SM.

Minor points:

Typos: "Examples are images an videos" "Examples are images and videos".

REPLY: corrected

The Arxiv paper cited [19] has been published in a peer reviewed journal and the latter should be referenced.

REPLY: corrected

Reviewer #2 (Remarks to the Author):

The paper is very clearly written, the code is provided, which is very good, my only suggestion to improve the manuscript is to somehow try to be more quantitative on the stability issue: currently the degradation of the reconstruction with time is only very qualitative (paragraph at the end of page 3). It would be nice to give more quantitative data such as correlation of the speckle over time and reconstruction capability. These sensitivity to bending and decorrelation have been studied quite extensively in ref 18 and 20 and it would be nice to see how this method compare.

REPLY: we agree with the referee that the issue of stability over a time is an interesting point. We note however that the main goal of this work was to report an approach that allows high quality reconstruction of images through a fibre with a novel ANN. Moreover, it is somewhat difficult to quantitatively compare the stability of our system with respect to previous work due to the lack of a consistent framework within which to do this. In order to partly address these issued and difficulties, we have looked in more detail at the stability of our own data. In the SM file we have now updated Figure 2 so as to include a series of measurements taken at various times (1, 16, 40, 52 hours). In Fig. 2e, 1st row, we show the measured speckle patterns and calculate the SSIM that measures the correlation between the images at various times compared to the 1st image ($t = 1$ h). As can be seen, the SSIM decreases gradually in time, indicating the slow decorrelation of the speckle pattern mentioned by the referee. The corresponding retrieved images, shown on the second row, exhibit a similar decrease in the SSIM. We believe that this is the first attempt to quantify speckle decorrelation with image quality and hopefully this will become a widespread approach so that comparisons can be made. Other measures of the speckle and image quality could of course be used but we would expect similar results, i.e. a linear relation between speckle correlation and final image quality. As mentioned in the SM, in future work we will indeed focus on deepening our understanding of this process and how to control it, e.g. by increasing the learning database so as to account for the speckle decorrelation. Similar considerations will apply also to bending. There is no quantitative measure at the moment for fibre bending: we believe that approach shown here and suggested by the referee is a promising platform for future studies in this direction.

REVIEWERS' COMMENTS:

Reviewer #1 (Remarks to the Author):

The authors clarified the questions raised point by point with clarity and addressed satisfactorily in the manuscript.

Reviewer #2 (Remarks to the Author):

I am ok with the answer, and I recommend the paper for publication.